# Effects of *Mustelid gammaherpesvirus 1* (MusGHV-1) Reactivation in European Badger (*Meles meles*) Genital Tracts on Reproductive Fitness

**DOI:** 10.3390/pathogens9090769

**Published:** 2020-09-20

**Authors:** Ming-shan Tsai, Ursula Fogarty, Andrew W. Byrne, James O’Keeffe, Chris Newman, David W. Macdonald, Christina D. Buesching

**Affiliations:** 1Wildlife Conservation Research Unit, Department of Zoology, University of Oxford, Recanati-Kaplan Centre, Abingdon Road, Tubney House, Tubney, Oxfordshire OX13 5QL, UK; chris.newman@zoo.ox.ac.uk (C.N.); david.macdonald@zoo.ox.ac.uk (D.W.M.); christina.buesching@lmh.ox.ac.uk (C.D.B.); 2Irish Equine Centre, Johnstown, Naas, Co. Kildare W91 RH93, Ireland; UFogarty@irishequinecentre.ie; 3One-Health Scientific Support Unit, Department of Agriculture, Agriculture House, Dublin 2 DO2 WK12, Ireland; AndrewW.Byrne@agriculture.gov.ie; 4Department of Agriculture, Agriculture House, Dublin 2 DO2 WK1, Ireland; james.okeeffe@agriculture.gov.ie; 5Centre for Veterinary Epidemiology and Risk Analysis, University College Dublin, Belfield, Dublin 4 D04 W6F6, Ireland; 6Cook’s Lake Farming Forestry and Wildlife Inc (Ecological Consultancy), Queens County, NS B0J 2H0, Canada

**Keywords:** gammaherpesvirus, badger, MusGHV-1, reproductive failure, genital tract, sexually transmittable infection (STI), sexually transmittable disease (STD), vertical transmission, reactivation, latency

## Abstract

Reactivation of latent Gammaherpesvirus in the genital tract can lead to reproductive failure in domestic animals. Nevertheless, this pathophysiology has not received formal study in wild mammals. High prevalence of *Mustelid gammaherpesvirus 1* (MusGHV-1) DNA detected in the genital tracts of European badgers (*Meles meles*) implies that this common pathogen may be a sexual transmitted infection. Here we used PCR to test MusGHV-1 DNA prevalence in genital swabs collected from 144 wild badgers in Ireland (71 males, 73 females) to investigate impacts on male fertility indicators (sperm abundance and testes weight) and female fecundity (current reproductive output). MusGHV-1 reactivation had a negative effect on female reproduction, but not on male fertility; however males had a higher risk of MusGHV-1 reactivation than females, especially during the late-winter mating season, and genital MusGHV-1 reactivation differed between age classes, where 3–5 year old adults had significantly lower reactivation rates than younger or older ones. Negative results in foetal tissues from MusGHV-1 positive mothers indicated that cross-placental transmission was unlikely. This study has broader implications for how wide-spread gammaherpesvirus infections could affect reproductive performance in wild Carnivora species.

## 1. Introduction

The herpesviridae comprise four subfamilies, the alphaherpesvirinae, betaherpesvirinae, gammaherpesvirinae [1], and the newly assigned deltaherpesvirinae [2], which can all maintain lifelong infection, and exhibit prolonged latency and repeated reactivation. Infection typically causes no, or only mild subpatent, disease [3], although serious symptoms may be observed in immuno-compromised individuals [4]. In most cases, reactivation and virus shedding are not associated with symptomatic disease [5], but even asymptomatic reactivation results in host infectiousness [5].

In general, the extent of clinical symptoms during the lytic stage of herpesvirus infection depends on both, innate and acquired host immunity [6]. Across species, stimuli associated with reactivation include physical or mental stress [7,8], trauma [9], hypoxia [10], coinfection with other pathogens [11], pregnancy [12], hormonal changes [13], and aging [14].

Primary herpesvirus infection, or reactivation from latency, during pregnancy can be serious, causing reproductive failure, including infertility, embryonic reabsorption, foetal abortion, preterm birth, stillbirth, poor neonatal condition, or neonatal death [15,16]. The mechanisms involved may be direct or indirect [15]. Direct herpesvirus-induced abortions are usually associated with viremia by primary infection or reactivation during pregnancy, leading to heavy viral loads in uterine vessels and cross-placental transmission to the foetus(es), resulting in reabsorption, abortion, or neonatal death. This has been demonstrated in alphaherpesviruses, such as *Canine herpesvirus 1* (CHV-1) in dogs [17], Bovine herpesvirus 1 (BHV-1) in cattle [18], *Equine herpesvirus 1* (EHV-1) and occasionally *Equine herpesvirus 4* (EHV-4) in horses [15,19], and *Cervid herpesvirus 2* (CHV-2) in reindeer [20]. In contrast, indirect abortigenic sequelae associated with herpesvirus reactivation usually involve secondary bacterial infection. This may occur because viral infection can decrease foetal protection from bacteria ascending the vaginal tract or peritoneal cavity [21], or they may change the maternal immune response to commensal bacteria and cause preterm birth [22]. This has been demonstrated for Gammaherpesvirinae, such as the *Murid herpesvirus 4* (MHV-4) in mice [23], *Bovine herpesvirus 4* (BHV-4) in cattle [24], and has been suggested for *Equine herpesvirus 2* and *5* (EHV-2 and EHV-5) in horses [25].

Despite these potential impacts on reproduction, herpesvirus reactivation and associated diseases have rarely been investigated in wildlife. In the Carnivora, reported herpesvirus infections belong mostly to the gammaherpesvirinae, and have been documented from casualties [26,27], epidemiological surveys [28,29,30], endangered species management [31,32,33], or incidentally from studies that were focused on other pathogens [31]. Here we report for the first time the results of a systematic screening study of wild European badgers (*Meles meles*; henceforth ‘badger’) from populations across the Republic of Ireland, investigating how *Mustelid gammaherpesvirus 1* (MusGHV-1) reactivation in genital tracts affects host reproductive performance.

Badgers are an informative wildlife model to study the effects of sexually transmittable infections (STIs), because they have a highly promiscuous, polygynandrous mating system characterized by repeated mounting behaviour [34], likely putting them at particular risk of contracting STIs [35]. Their social system is highly flexible and depends largely on resource availability, ranging from solitary or pair-living to living in social groups comprising up to 29 adults [36]. They are seasonal breeders, and produce one litter of one to a maximum of five cubs annually between mid-January to end of February after 6 weeks of gestation [37]. In Ireland and the UK, 93% of litters comprise less than three viable cubs [38], with an average litter size of 2.33–2.5 (min = 1, max = 4, Ireland: [37,39]). The primary mating season occurs during the postpartum oestrus (UK/Ireland: from mid-January to end of April) with possible additional conceptive matings occurring through the summer months into October [40]. Badger reproduction is characterized by delayed implantation (i.e., the fertilized ova develop to the blastocyst stadium, but then remain in diapause for up to 11 months until implantation in mid-late December) and superfoetation (i.e., females can continue to ovulate and conceive during delayed implantation) [40]. Nevertheless, in areas with high population density, badger fecundity is low (e.g., Southern England [41]): although conception rate can reach 90% [41], Roper [42] estimated that on average only 14% of blastocysts result in viable cubs, of which a further 60% die pre-emergence [42]. In contrast, in lower density populations, female badgers have lower conception rate (66%), but their abortion rate is also lower (estimated in Ireland as <1% in January from post-mortem studies: [39]), corroborating previous evidence [34] that prepartum reproductive success may be largely affected by population density. Previous studies have linked this high reproductive failure rate in part to intra-sexual reproductive suppression [43], somatic vs. reproductive fitness trade-offs [44], infanticide in Southern England [42] (but not in Ireland [39]), and high neonatal cub mortality due to coccidiosis [45]; but no single suggested cause can fully explain the observed phenomenon [42]. Surprisingly, the potential for this reduced fecundity, infertility, or abortigenic disease to be linked to STIs [46], has been largely ignored in badgers (but see [47]).

The *Mustelid gammaherpesvirus* 1 (MusGHV-1), a member of the genus Percavirus, was first isolated from lung tissue of a wild badger in 1996 without obvious association to disease [48]. It is highly prevalent amongst wild badgers in the UK [49] and Ireland [50], where MusGHV-1 DNA isolation from blood shows near 100% infection rate at some localities. Nevertheless, unlike alphaherpesviruses that reside in neuron cells during latency, gammaherpesviruses reside in B-lymphocytes (EBV, MHV-4, EHV-5) [4,51,52] or monocytes (BHV-4) [53], and therefore it is not possible to differentiate latency from reactivation when isolating viral DNA from whole blood samples. Confirming sexual transmission in a free-ranging population without using manipulative experiments is extremely challenging, but indirect evidence permits inference of horizontal and/or vertical transmission, for example, by isolating the pathogen DNA using PCR from swabs of genital tracts, placenta tissues, or foetal tissues. A population-wide screening survey revealed a reactivation rate of 55% in genital tract mucosae of badgers in a high density population in the UK [47], indicating a high likelihood of sexual transmission. This same study also linked reactivation of genital MusGHV-1 to reduced future reproductive success in female badgers [47], but did not investigate the factors contributing to herpesvirus reactivation. Here, we use badger samples collected post-mortem from across the Republic of Ireland under a national culling programme for the management of bovine tuberculosis to investigate:Patterns of MusGHV-1 reactivation in badger genital tracts by comparing prevalence among females culled at the start of the primary mating season with those culled at the end (where many females are pregnant at the beginning, but all will have given birth by the end);Differences in MusGHV-1 reactivation between males and females, and between different age groups;Effects of MusGHV-1 reactivation on female reproductive success, male testes weight, and sperm abundance;The potential for vertical transmission of MusGHV-1 to the foetus via the placenta during gestation, or via the maternal vaginal tract during parturition; andThe potential for horizontal transmission through semen.

We discuss our results in the framework of potential impacts of MusGHV-1 infection on badger population dynamics and reproductive health and in the broader context of carnivore conservation.

## 2. Results

### 2.1. Risk Factors of MusGHV-1 Reactivation in Badgers

Overall, prevalence of MusGHV-1 in genital swabs was 61.8% (95%CI: 53.7–69.3%, 89/144 swabs). The trimmed 23 sequences (691 bps) were 100% identical to each other and to the previously reported MusGHV-1 sequence (AF275657) isolated from lung tissue cultures of a female badger in South England in 1996 [48].

In winter (i.e., for which we had data on both, sex and age), sex and age were significant risk factors for genital MusGHV-1 reactivation (see results of multivariable logistic regression analysis in Table 1, area under receiver operating characteristic curve (AUROC) = 0.74. If AUROC equals 0.5 means the model has no accuracy of diagnostic; while AUROC equals 1 means the model has perfect accuracy). At the beginning of the mating season, MusGHV-1 reactivation in genital tract was 6.7 times more likely to happen in males (82.5%, 33/40) than in females (47.5%, 19/40), and 4.7 times more likely in young adults (72.4%, 21/29) than in middle-age adults (50%, 10/20) (prevalence in Table 2 and odds ratio in Table 1). Particularly, middle-aged males had a 6-fold higher prevalence of genital MusGHV-1 than did females in the same age category (75%, 9/12 vs. 12.5%, 1/8, respectively). If yearlings were excluded (due to very low sample size), our data suggested no difference of prevalence between age groups was found in males, but in females, prevalence in middle aged individuals (12.5%, 1/8) tended to be lower than in young (58.8%, 10/17) or in old (63.6%, 7/11) adults, although this association did not reach significance (Chi-squared = 5.8476, df = 3, *p* = 0.054).

Univariate follow-up analyses for both seasons, however, showed that males had a significantly higher risk of MusGHV-1 reactivation than females only in winter, but not in spring (winter prevalence in males = 82.5% (33/40), females = 47.5% (19/40), Chi-squared = 9.2857, df = 1, *p*-value = 0.002; spring prevalence in males = 64.5% (20/31), females = 51.5% (17/33); Chi-squared = 0.6388, df = 1, *p*-value = 0.424, see Table 2 for full results).

### 2.2. Interactions of Genital MusGHV-1 Reactivation with Body Weight

There was no overall body weight difference between MusGHV-1 positive and negative individuals, with the exception of females in winter. As pregnancy is energetically costly, we further separated females into pregnant and nonpregnant, and evidenced that MusGHV-1 was associated with lighter bodyweight only in pregnant females, but not for those that had already given birth or had not been pregnant; nevertheless, this effect was not statistically significant, perhaps due to the small sample sizes (*t* = 1.8691, df = 4.8914, *p*-value = 0.122; *t*-test) (Figure 1).

### 2.3. Effects of Genital MusGHV-1 Reactivation on Reproductive Fitness

Relative sperm abundance ranged from 0 to 160.38 × 10^5^/mL with an average of 43.89 × 10^5^ spermatozoa/mL (±40.18 × 10^5^) of equally diluted semen collected from 39 males. However, there was no significant effect of genital MusGHV-1 reactivation on relative testicular weight (*p* = 0.82), but a trend—albeit not significant (*p* = 0.085)—for positive males to have higher sperm abundance (Figure 2).

In the winter dataset (2020), 71.8% (28/39) of females implanted successfully while 61.5% (24/39) reproduced successfully (i.e., still pregnant or recently pregnant with fresh placental scars at time of culling) with an average litter size of 2.46 ± 0.72 (min = 1, max = 4). In the spring dataset (2019), 51.6% females (16/31) had signs of recent pregnancy with an average of 2.41 ± 0.9 (min = 1, max = 4) placental scars (representing most recent litter sizes); blastocysts were present in the uterus of 52.2% (12/23) of adult females with an average of 2.08 ± 0.63 blastocysts (min = 1, max = 3, *n* = 12) per pregnant female. There was no seasonal/annual difference in reproductive fitness between the two sampling periods (Chi-square = 0.3486, df = 1, *p*-value = 0.555).

Notably, females with genital MusGHV-1 reactivation (see Table 3) had a significantly lower rate of successful pregnancy (41.2%, 14/34) than MusGHV-1 negative females (72.2%, 26/36) (odds ratio = 0.269, 95%CI = 0.1–0.73, Chi-square = 5.672, df = 1, *p*-value = 0.017). This indicates that MusGHV-1 reactivation had a negative effect on female reproductive fitness, although when separated by season this trend was only significant in winter but not in spring (Table 4). No significant association was found between MusGHV-1 genital reactivation and the presence of blastocysts in spring (Appendix A).

### 2.4. Potential for Horizontal and Vertical Transmission

One (20%) seminal plasma sample from genital MusGHV-1 negative males tested positive for MusGHV-1, but interestingly none of the five samples from genital MusGHV-1 positive males contained cell-free MusGHV-1 DNA. From 18 well-preserved pregnant females, we dissected a total of 43 foetuses (partial dissection of 33 and full dissection of 10 foetuses), which all appeared to be healthy and normally developed, irrespective of the mother’s herpes infection status. Three out of 10 placentas tested were positive for MusGHV-1 DNA, (one from a genital MusGHV-1 negative mother; two from mothers with partial foetal loss—one tested positive for genital MusGHV-1, one negative); however, no MusGHV-1 DNA was detected in any foetal tissues.

### 2.5. Abnormalities in Female Reproductive Tracts Associated with Genital MusGHV

In 25% (7/28) of females (three young and four old adults) that were pregnant or showed signs of recent pregnancy, we recorded possible abortions (enlarged uterus but no foetuses/placental scar(s) present, but with signs of reabsorption, such as dark, amorphous mass of dead cells in the enlarged uterus [41]) of entire 14.3% (4/28) or partial (10.7%; 3/28) litters. All four females (three were >5 years old, one was <3 years old) with complete litter loss showed no sign of lactation and tested positive for genital MusGHV-1 reactivation Furthermore, one of the four pregnant MusGHV-1 positive females had reddened cyst-like gross lesions surrounding the cervix; one had submucosal haemorrhage around the cervix, and one had the remains of an absorbed foetus (in addition to two healthy ones) in her uterus; only one showed no abnormalities in her reproductive tract.

We also observed that 48.38% females (30/62, 40 from winter and 22 from spring) and 52.5% males (21/40, all from winter) displayed macroscopic lesions that resembled lymphoid hyperplasia on the epithelium of the lower genital tracts or around the cervix in females, and the mucosal epithelium of the outer urethra in males. These gross lesions took the form of vesicle-like (Figure 3A), congestive and oedemic (Figure 3B), or pigmented dotted scars (Figure 3C), but were not always associated with genital MusGHV-1 reactivation (Appendix A). We only tested one lesion tissue (Figure 3B) for MusGHV-1 and the result was positive, but the genital swab of this specific animal tested negative.

Histopathological examination confirmed that all seven samples from animals with gross lesions investigated (females: one cervix and four inner genital tract tissue samples from females; males: two inner genital tract tissue samples) showed inflammation, with five out of the six genital tract tissues exhibiting epithelial hyperplasia (Appendix A). Among these, only one male mucosal tissue collected from the base of the penis had intra-nuclear inclusion bodies in the epithelium, likely indicating viral infection (Figure 4).

Genital discharge was present in 74.2% (23/31) of females during spring, and 7.5% (3/40) of females and none of the males (0/40) in winter, but was not associated with MusGHV-1 reactivation. A cervical mucus plug (CMP) was observed in 44% (8/18) of the pregnant females, none of which tested positive for genital MusGHV-1 reactivation (compared to 40% genital MusGHV-1 prevalence in the remaining 10 pregnant females without a CMP).

## 3. Discussion

Genital MusGHV-1 reactivation rates in our Irish dataset are comparable to those previously reported in an English badger population [47]. Nevertheless, the overall badger MusGHV-1 reactivation rates of 61.8% found in this study and of 50.6% (excluding cubs) in Kent et al. [47] at a near 100% prevalence in blood [49] are higher than gammaherpesvirus reactivation rates reported in other species: human EBV (22–90% seropositive rate) has been reported with viral DNA isolation from 8–28% from cervical fluids and semen samples [54]; horse EHV-5 DNA has been found in 14.7% of uterine flushing samples and 2.6% of genital swabs; horse EHV-2 (79.7% seroprevalence [55]): 1.6% from uterine flushing; 2.3% from genital swabs [25,56]; reindeer CvHV-2 DNA has been found in 24% of vaginal swabs (unspecified seroprevalence) [20]; cattle BHV-4 (96.9% seroprevalence) DNA in 48.5% of uterine swabs and 51.5% of vaginal swabs [57]. The exceptionally high prevalence of MusGHV-1 DNA in badger genital tracts further strengthens that sexual transmission may be an important route in gammaherpesvirus transmission comparable to human KSHV [58] and murine MHV-4 [13].

Nevertheless, MusGHV-1 reactivation rates fluctuated over a badger’s lifetime: middle aged (3–5 years old) individuals had lower reactivation rates than young (2–3 years old) and over 5 year old animals, and all MusGHV-1 positive pregnant female badgers belonged to the old age group. This mirrors results from humans and some domestic animals where primary infection occurs predominantly during infancy [59], with initial reactivation in mucosae near the entry portal [60] and repeated reactivation throughout life [51], which increases with aging due to declining immune-competency [14].

Overall, we found no difference in the body weights of genital MusGHV-1 positive and negative individuals suggesting that MusGHV-1 reactivation does not affect—nor was triggered by—host body condition, mirroring results from in England [47]. We did, however, identify a nonsignificant trend in pregnant females, where genital MusGHV-1 positive individuals had lighter body weights than those that tested negative (Figure 1), indicating that during pregnancy, reactivation may lead to decreased body condition [61], or—conversely—that reduced body condition during pregnancy may promote MusGHV-1 reactivation [62].

### 3.1. Sex-Specific Seasonal Patterns in Genital MusGHV-1 Reactivation

Our study shows that male badgers are significantly more at risk of genital MusGHV-1 reactivation than females, particularly at the beginning of postpartum mating season (end of January and early February). Latent sexually transmittable pathogens likely evolved to be reactivated during their host’s mating season to maximize host-to-host transmission [63]; a pattern well-established for cyprinid herpesvirus CyHV-3 in wild common carp [64]. The main mating season in badgers coincides with the postpartum oestrus (in the UK and Ireland typically in the middle of February: [37,65], but males start their “rutting behaviour” several weeks earlier from the middle of January [66], coinciding with a rise in their testosterone levels [67]. Testosterone, however, acts as an immune-suppressant [68] and may thus be partly responsible for the corresponding peak in male MusGHV-1 prevalence rates, mirroring results in domestic cats where males also have a higher risk than females to be infected with the feline gammaherpesvirus FcaGHV-1 [33,69], and entire males are at higher risk of infection with FcaGHV-1 than neutered males [70]. Female badger MusGHV-1 reactivation rates in the genital tract, in contrast, are higher in spring at the end of the main mating season than at its start in January. This could be due to the physiological stress linked to late-term pregnancy, birthing, lactation [71], and postpartum mating [72], but may also arise due to (re-)infection through mating with males shedding MusGHV-1 in their genital tract (e.g., equid alphaherpesvirus 3 reported in seropositive mares re-shed after mating with virus-shedding stallions [73]), where our results show that over 80% MusGHV-1 prevalence in genital tracts of male badgers, and 10% semen contained viral DNA.

### 3.2. Effects of Genital MusGHV-1 Reactivation on Reproductive Fitness

Although reproductive failure [41] and premature cessation of lactation [39] are often reported in badgers, specific studies investigating potential links to reproductive tract diseases are lacking. MusGHV-1 reactivation did not decrease male sperm production; in fact, there was a nonsignificant trend for MusGHV-1 positive males to have a higher sperm count than MusGHV-1 negative ones (Figure 2A), although we found no difference in testicular weight connected to genital herpes (Figure 2B). This suggests that, in males, genital MusGHV-1 reactivation does not cause sterility, although effects on sperm morphology, motility, and viability remain unknown (see also humans: [74]).

In females, in contrast, genital MusGHV-1 reactivation has a negative effect on reproductive fitness, manifesting in failure to carry pregnancy to term as seen in in other species [15]. The abortion rate (entire litter: 14.3% and partial 10.7%) confirmed in this study is considerably higher than the <1% previously reported in Irish badgers [37,39]. This is likely due to the concentrated sampling immediately before the main birthing period in this study, which appears to be more likely to observe abortions through gross examination, but comparable to rates reported from England (complete litter loss: 20–33% [42], partial loss: 10–14.3% [42]). In winter, genital MusGHV-1 prevalence rates were significantly higher in females that did not reproduce than in those carrying healthy foetuses or having carried their pregnancy to term. All four females with signs of spontaneous abortion of their full litters, as well as one out of the three females with late partial abortion tested MusGHV-1 positive. This further substantiates the negative association of genital MusGHV-1 reactivation with badger female reproductive fitness. In addition, 4/5 of these were old, potentially indicating that aging in in combination with increasing stress when approaching parturition [75] may favour latent MusGHV-1 reactivation, thus increasing the likelihood of abortion [76].

In domestic animals, gammaherpesvirus-associated reproductive failure has frequently been shown to be associated with bacterial coinfection [19,20,21,22]. During pregnancy, cervical mucous plugs (CMPs) likely form a physical and immunological barrier to prevent microbes ascending into the uterus from the lower reproductive tract (e.g., humans: [77]; mares: [78]). While we did not investigate potential differences in the genital microbiome community structure, all pregnant females with a CMP tested MusGHV-1 negative, compared to a 40% prevalence of genital MusGHV-1 reactivation in females without a CMP, suggesting that reproductive success may be affected by complex relationships between MusGHV-1 reactivation, bacterial microbiome dynamics, and host immunity [79].

### 3.3. Genital Lesions

Both sexes commonly had gross lesions in their genital tracts resembling the lymphoid hyperplasia and hyperaemia of the vaginal mucosa characteristic of female dogs infected with CHV-1 [80], and similar to the progress of equine coital exanthema caused to horse genitalia by EHV-3 infection [81,82]. In the case of badgers, however, we did not find a significant correlation between the presence of gross genital lesions and MusGHV-1 reactivation, nor with female reproductive success, although intranuclear inclusion bodies—an indicator of herpesviral lytic infection [83]—were found during histopathological examination in epithelial cells of the genital mucosa of one male with epithelial hyperplasia. This may be due to lytic infection and virion-shedding not necessarily being synchronized with the formation of lesions, as reported from experimental CvHV-2 reactivation in reindeer [20].

### 3.4. Vertical Transmission of MusGHV-1

Our findings suggest that vertical transmission of MusGHV-1 through the placenta is possible but unlikely. This corroborates other studies on gammaherpesvirus transmission (e.g., MHV-4 transmission in mice [13]; EBV in humans [82]), but contradicts a study of BHV-4 transmission in cattle, where intrauterine infection was found in 4/7 stillbirth foetuses [76]. Nevertheless, primary infection in foetuses may likely happen during parturition, or through close postnatal contact with the mother, explaining results from a previous study that found 75% 3-month old prepubescent cubs presenting positive for genital MusGHV-1 [47].

### 3.5. The Significance of Gammaherpesviruses as Drivers of Wildlife Population Dynamics

Gammaherpesviruses are of increasing concern in humans [51,58] and animals (livestock: [15], pets: [84], wild [28,31,48,69], and captive wildlife populations: [26]). EBV is prevalent in 90% of the human population, and is associated with diseases such as infectious mononucleosis, childhood leukaemia from maternal infection and, more rarely, lymphoma [51,85,86]. Bovine herpesvirus 4 is associated with abortion in cattle [15], and the widespread equid herpesvirus 2 and 5 are associated with rare diseases such as fatal equine multinodular pulmonary fibrosis [52], and there is increasing evidence of their connection to other reproductive disorders [25]. Most recently, infections by a novel gammaherpesvirus *Leporid gammaherpesvirus 5* (LeHV-5) were identified in wild Iberian hares (*Lepus granatensis*) with genital and skin lesions, and will likely intensify the recent population decline [87].

Although gammaherpesvirus associated diseases have rarely been reported in wild animals, this observation might be due to the difficulty of diagnosis in the field, and to asymptomatic infection, especially for abortigenic diseases. However, reports of novel herpesviruses in wildlife have strengthened the assumption of ubiquity across the animal kingdom. Broad surveys of occurrence and in-depth research on associated diseases in wildlife are warranted to enhance understanding of gammaherpesvirus effects on epidemiology, pathogenicity, evolution, and disease ecology. As theory predicts [88,89] and empirical studies evidence [90,91], STI-induced sterility can constitute an important driver of population dynamics, and superficially benign infections without clinical manifestation may in fact drive already endangered wildlife populations to extinction.

## 4. Materials and Methods

### 4.1. Source of Post-Mortem Badgers

Samples were collected at the Irish Equine Centre (IEC) between 23rd and 26th April 2019 (lactation/end of primary mating season) and 3rd–7th February 2020 (end of gestation/onset of the primary postpartum mating season) post-mortem from 144 badgers (71 males and 73 females) from across the Republic of Ireland culled under licence by the National Parks and Wildlife Service, with powers conferred by the Wildlife Acts 1976 to 2010 in the context of a national removal scheme to control bovine tuberculosis led by the Department of Agriculture, Food and the Marine (DAFM) [92] (see Table 2 for details). Carcasses were preserved in a cold room at 0–2 °C as soon as possible after death and were examined between 1 and 10 days after death (average = 4.58 ± 1.95 days, median = 4 days). For each badger, culling site, sex, weight, and approximate age group (estimated from tooth wear: [93] and categorized as cubs: <1 year old; yearling: <2 years old; young adult: 2–3 years old; middle-aged adult: 3–5 years old; old adult: >5 years old) were recorded. All badgers were examined for genital discharge and lesions in the reproductive system.

### 4.2. Assessment of Reproductive Fitness

In females, appearance and interior of the uterus were examined and reproductive status was classified as follows: pregnant (carrying developing foetus(es)), recently pregnant (enlarged and thickened uterus with dark pigmented placental scars and mammary development/lactation), recently pregnant but aborted (enlarged and thickened uterus with no or pale placental scars, or no mammary development), not pregnant (determined by opaque round uterine appearance, absence of placental scars and small nonelongated teats; or thin translucent uterus with no evidence of uterine thickening, placental scars, or mammary development) [39]. Uterine flushing with 0.9% saline solution and following examination of the flush exudate under a dissecting microscope at 4× magnification were used to determine recent, successful mating (i.e., presence and number of blastocysts in uterus). Litter size was determined through direct counts of foetuses if the female was pregnant, or placental scar counts if it had already given birth [94]. Lactational status was determined peripartum in winter by gross and histological examination of the mammary gland tissues, and postpartum in spring by mammary development and presence of milk [39].

In males, one testis was collected using sterile scissors, and after removing all attached fat, muscles and blood vessels, was weighed in g using a precision balance [67]. The epididymis was then detached and placed on a sterile petri dish, before being macerated and mixed with 2.5 mL 1× PBS to release the semen. The diluted semen was transferred to a sterile 2 mL microcentrifuge tube and stored at 4 °C before processing. Relative abundance of sperm was determined using a haemocytometer (1:10 diluted by 1× PBS) under a microscope at 40× magnification [95].

### 4.3. Assessing Potential for Vertical and Horizontal Transmission of MusGHV-1

To investigate, if MusGHV-1 can be transmitted vertically prebirth (transplacentally), we collected maternal placentas and foetal brains, lungs, spleens, and livers from eight pregnant females (four females that tested MusGHV-1 positive in genital swabs and four females that tested negative in genital swabs—i.e., still with potential for latent infection or reactivation in other body-sites). To investigate if horizontal transmission through cell-free MusGHV-1 DNA in semen is plausible, we selected diluted semen samples from 10 males (five genital swab MusGHV-1 positive and five genital swab-negative control samples to elucidate if potential cell-free viral DNA originates from sperm or from epithelial cells in the genital tract), and then collected seminal plasma from the supernatant after centrifugation at 1500 rpm for 10 min. All swabs, tissue, and semen samples were frozen immediately and stored at −20 °C before being shipped frozen and packaged in dry ice to the University of Oxford for DNA extraction and PCR.

### 4.4. Histopathological Examination

Mammary gland tissues from all females with recent pregnancy but uncertain lactation status were collected to confirm lactation by histological examination. Representative lower genital tract tissue samples with lesions were collected from five females and two males (according to availability) and fixed in 10% neutral buffered formalin. These formalin-fixed tissues were trimmed, dehydrated through a graded ethanol series, and then infiltrated with paraffin. These embedded tissues were then sliced at 4 μm and stained with haematoxylin and eosin (H&E) for microscopic examination.

### 4.5. MusGHV-1 DNA Detection in Genital Tracts, Semen, and Tissues

To determine MusGHV-1 reactivation in badger genital tracts (i.e., presence of cell-free MusGHV DNA), lower genital tracts were swabbed with sterile cotton tops with wooden shafts (Dynarex, New York, USA) to secure virions without damaging the epithelium. Any swabs contaminated with blood (from culling/prior dissection) were centrifuged at 1500 rpm for 10 min to eliminate circulating white blood cells that could contain latent MusGHV-1. Samples were then placed in empty 2 mL sterile microcentrifuge tubes and stored at −20 °C.

A ca. 25 mg subsample of each tissue was taken, and each swab was reconstituted with 400 µL sterile doubly distilled water, vortexed gently at room temperature for 10 min, before decanting a 200 µL aliquot from the supernatant. DNA was processed using DNeasy Blood and Tissue Kit (Qiagen), following the manufacturer’s instructions. Purified DNA was then eluted in 100 µL of the buffer provided.

Purified DNA was screened for MusGHV-1 using a specific primer pair (former primer: 5′-CCA AGC AGT GCA TAG GAG GT-3′; reverse primer: 5′-TGG ACT TCT CCA ACA TGC GTC GCC CTT C-3′) modified from King et al. 2004 [50], targeting 771 base pairs of a partial DNA polymerase gene. For each reaction, a total of 20 µL PCR solution was mixed with 10 µL HotStartTaq Master Mix (Qiagen) containing 1 unit of HotStartTaq DNA Polymerase, 12 µM of MgCl_2_, and 1.6 µM of each dNTP), 0.5 µM of front and reverse primer, 2 µL Coralload gel loading dye, and 5 µL DNA template. Amplification conditions were kept at 95 °C for 5 min to activate DNA polymerase, followed by 45 cycles of denaturation at 95 °C for 45 s, primer annealing at 60 °C for 45 s, and chain elongation at 72 °C for 1 min, followed by a final extension at 72 °C for 10 min. Finally, the PCR products were loaded in 2% agarose gel to check amplification results under UV light. Representative PCR products with positive results were genotyped by Sanger sequencing (Zoology Sequencing Facility, University of Oxford), and one representative sequence was submitted to the NCBI Genbank under accession number MT332102. Trimmed sequences were compared to previously published sequences on GenBank by the online NCBI Blast service.

### 4.6. Statistical Analysis

All sample details and variables are shown in Appendix A. Statistical analyses were performed using *R* Studio software (version 1.21335). Prevalence of MusGHV-1 was calculated as the percentage of PCR positive cases over the total number of samples examined, and 95% upper and lower confidence intervals were calculated using the Wilson method [96]. Univariate analysis using Chi-square tests with Yates continuity correction and Fisher’s exact test (if any column of the contingency table was less than 5) were used to investigate if age, sex, or age groups were risk factors for genital MusGHV-1 reactivation in badgers, and whether genital MusGHV-1 reactivation affects female reproductive fitness. To investigate whether there was an interaction between MusGHV-1 reactivation and badger weight, herpes-positive and negative captures of each sex, in each culling period, were compared separately with *t*-tests to standardize for sexual and seasonal weight differences [39]. To investigate effects of genital MusGHV-1 on testes weight and relative sperm concentration, herpes-positive and negative males were compared using *t*-tests. Logistic regression multivariable analysis and odds ratio were used to estimate the size effect of different risk factors on MusGHV-1 reactivation, and the area under receiver operating characteristic curve (AUROC) was calculated for diagnostic ability of the final model [97]. Due to data availability, age group analysis was only applied to samples collected in winter, and yearlings were excluded (due to very low sample sizes). We are currently working on a different (*n* = 251) dataset from a UK population, where we show that age has strong correlation to MusGHV-1 prevalence (quadratic term, adjusted R^2^ = 0.602, *p* < 0.001). We use adjusted R^2^ as effect size, predictor = 1, significance level = 0.05 to calculate the minimum size required to reach 0.8 statistical power [98], and determined that *n* = 8 has sufficient statistical power to reach a valid statistical result.

## Figures and Tables

**Figure 1 pathogens-09-00769-f001:**
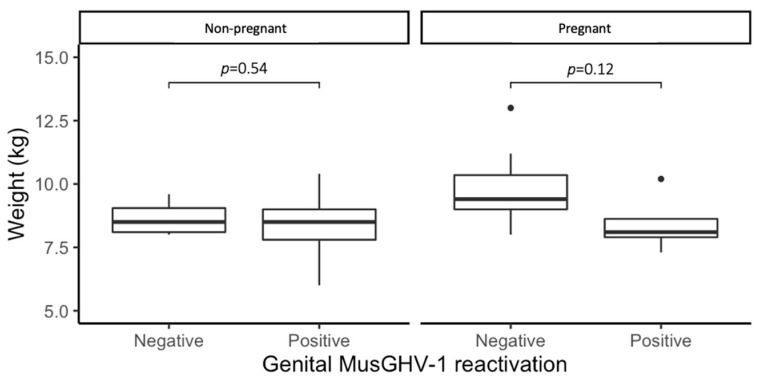
Weight comparison using *t*-test between pregnant and nonpregnant females. The black dots are referred to outliers.

**Figure 2 pathogens-09-00769-f002:**
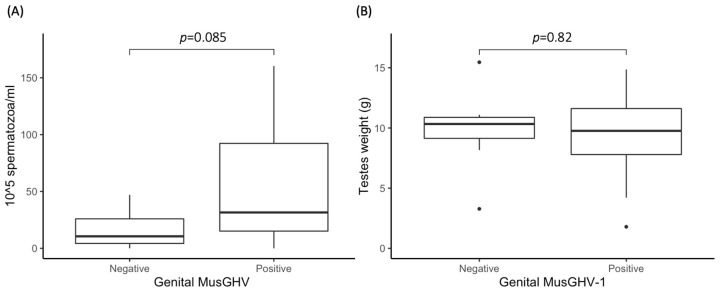
Comparison of sperm concentration (**A**) and testes weight (**B**) between genital MusGHV-1 positive and negative males. The black dots are referred to outliers.

**Figure 3 pathogens-09-00769-f003:**
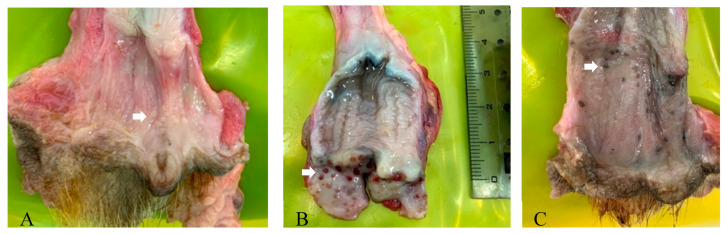
Lesions in the lower genital tract (possibly lymphoid hyperplasia); differences in appearance may reflect different stages of infection (**A**–**C**).

**Figure 4 pathogens-09-00769-f004:**
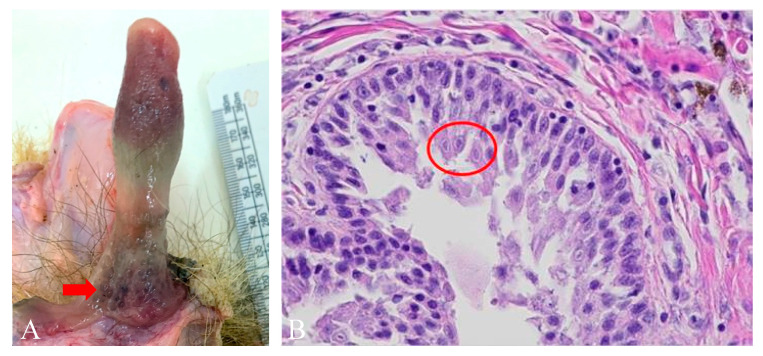
Image of gross lesion (**A**) found on preputial mucosa located at the base of the penis (arrows) and histology (**B**) showing intranuclear inclusion bodies (circles) in the apical side of epithelium.

**Table 1 pathogens-09-00769-t001:** Final model of multivariable logistic regression to identify risk factors for genital *Mustelid gammaherpesvirus 1* (MusGHV-1) reactivation (function = glm(MusGHV ~ Sex + Age, family = binomial), degrees of freedom = 68, AIC = 83.66) of winter samples with yearlings (*n* = 2) being excluded. Area under ROC = 0.74 in the final model.

Group	Estimate	Standard Error	*z* Value	Adjusted OR	95% CI of OR	*p* Value
(Intercept)	−1.1697	0.6466	−1.809	0.31	0.09–1.1	0.07
**Sex (vs. Female)**						
Male	1.9023	0.6442	2.953	6.7	1.9–23.67	0.003
**Age group**						
Young adults	1.5527	0.7243	2.144	4.72	1.14–19.53	0.032
Old adults	1.3507	0.7671	1.761	3.86	0.86–17.36	0.078

**Table 2 pathogens-09-00769-t002:** Univariate analysis of risk factors affecting MusGHV-1 DNA prevalence in badger genital tracts.

	Positive *n*	Total *n*	Prevalence	95% CI	Chi-Square Test
**Season**					
Winter	52	80	65%	54.1–74.5%	X^2^ = 0.503, df = 1, *p* = 0.478
Spring	37	64	57.80%	45.6–69.1%	
**Sex**					
Male	53	71	74.60%	63.4–83.3%	X^2^ = 8.741, df = 1, *p* = 0.003
Female	36	73	49.30%	38.2–60.5%	
**Age group ^a,b^**					
Yearling	2	2	100%	34.2–100%	X^2^ = 2.559, df = 1, *p* = 0.11 (Young:Middle aged)
Young	21	29	72.40%	54.3–85.3%	X^2^ = 0.034, df = 1, *p* = 0.854 (Young:Old)
Middle aged	10	20	50%	29.9–70.1%	X^2^ = 1.667, df = 1, *p* = 0.197 (Middle aged:Old)
Old	14	20	70%	48.1–85.4%	
**Season:Sex**					
Winter:Male	33	40	82.50%	68.1–91.3%	X^2^ = 9.2857, df = 1, *p* = 0.002
Winter:Female	19	40	47.50%	32.9–62.5%	
Spring:Male	20	31	64.50%	46.9–78.9%	X^2^ = 0.6388, df = 1, *p* = 0.424
Spring:Female	17	33	51.50%	35.2–67.5%	
**Sex:Age group ^a,b^**					
Male:Yearling	1	1	100%	20.7–100%	*p* = 0.59 (Young:Middle aged)
Male:Young	11	12	91.60%	64.6–98.5%	*p* = 0.553 (Young:Old)
Male:Middle aged	9	12	75%	46.8–91.1%	*p* = 1 (Middle aged:Old)
Male:Old	7	9	77.80%	45.3–93.7%	
Female:Yearling	1	1	100%	20.7–100%	*p* = 0.042 (Young:Middle aged)
Female:Young	10	17	58.80%	36–78.4%	*p* = 1 (Young:Aged)
Female:Middle aged	1	8	12.50%	2.2–47.1%	*p* = 0.059 (Middle age:Old)
Female:Old	7	11	63.60%	35.4–84.8%	

^a^: Yearlings are not included in the statistic calculation due to small sample size. ^b^: Only winter samples as age group data was not obtained in spring samples.

**Table 3 pathogens-09-00769-t003:** Female reproductive status and genital MusGHV-1 prevalence in European badgers separated by seasons and age groups (only in winter).

Reproductive Status	Spring				Winter				Young		Middle Aged		Old	
	(+)	Total	%	95% CI	(+)	Total	%	95% CI	(+)	(-)	(+)	(-)	(+)	(-)
Not pregnant	11	17	64.7%	41.3–82.7%	8	12	66.7%	39.1–86.2%	6	1	1	3	-	-
Pregnant ^a^	-	-	-	-	4	18	22.2%	9.0–45.2%	0	5	0	3	4	3
Recently pregnant	7	16	43.8%	23.1–66.8%	3	6	50.0%	18.8–81.2%	3	1	0	1	0	1
Recently pregnant, aborted	-	-	-	-	4	4	100.0%	51.0–100%	1	0	-	-	3	0

^a^: Three of the pregnant females showed signs of partial litter loss.

**Table 4 pathogens-09-00769-t004:** Risk factor analysis of female reproductive fitness.

	Pregnant *n* ^a^	Total *n* ^b^	Rate	95% CI	Chi-Square Test
**Season**					
Winter	24	39	61.50%	45.9–75.1%	X^2^ = 0.3486, df = 1, *p* = 0.555
Spring	16	31	51.60%	34.8–68%	
**Genital MusGHV-1**					
Positive	14	34	41.20%	26.4–57.8%	X^2^ = 5.672, df = 1, *p* = 0.017
Negative	26	36	72.20%	56–84.2%	
**Season:Genital MusGHV-1**					
Winter:Positive	7	18	38.90%	20.3–61.4%	X^2^ = 4.0032, df = 1, *p* = 0.045
Winter:Negative	17	21	81%	60–92.3%	
Spring:Positive	7	16	43.80%	23.1–66.8%	X^2^ = 0.2972, df = 1, *p* = 0.586
Spring:Negative	9	15	60%	35.7–80.2%	

^a^: This includes females that were pregnant or successfully gave birth (i.e., had fresh placental scars). ^b^: This excludes two prepubescent yearlings from spring and one from winter.

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
