# Peer review of "Effects of Mustelid gammaherpesvirus 1 (MusGHV-1) Reactivation in European Badger (Meles meles) Genital Tracts on Reproductive Fitness"

_pathogens, 2020, doi:10.3390/pathogens9090769_

Round 1

Reviewer 1 Report

The manuscript “Effects of Mustelid Gammaherpesvirus 1 (MusGHV-1) Reactivation in Badger (Meles meles) genital tracts on Reproductive Fitness” (pathogens-905329) by Ming-shan Tsai and collaboratorsreports the investigation of Mustelid gammaherpesvirus 1 (MusGHV-1) reactivation in 144 wild badgers (Meles meles) by testing genital swabs by PCR. The authors showed that MusGHV-1 impacts negatively on female reproduction but not on male fertility (inferred by testicular weight and sperm concentration), despite males having a higher risk of MusGHV-1 reactivation than females, especially during the late-winter mating season. They also showed that genital MusGHV-1 reactivation differs between age classes, with adults having significantly lower reactivation rates than younger or older individuals, similarly to what is observed with other herpesviruses. The results also showed that cross-placental transmission is unlikely.

General comments

The data produced is relevant to the knowledge of the pathogens that affect this wild carnivore species for which apart from Mycobacterium bovisvery few information is available. Regarding the scientific content, the manuscript is generally well written and detailed. However, several typing corrections are necessary concerning parentheses that were not closed (e.g. line 290), wrong colon punctuations (e.g. line 50), incomplete sentences (line 267 (e.g.,)), trash text “but see below” and an “Error…” (lines 92 and 214), etc.

Key words

Include also “latency” and “reactivation”.

Abstract

Line 23. Add the geographical origin of the samples since it is not in the title of the manuscript.

Introduction

Line 37. The proposed fourth subfamily of mammalian herpesviruses (Deltaherpesvirinae that includes ElHV-1is not mentioned by the authors. Despite the recommendation is yet to be ratified by the International Committee on Taxonomy of viruses, the new subfamily has been largely referred in the literature.

Line 97. As it is written, it seems that the sites of latent infection include only the hematopoietic (monocytes, B cells and T cells) and not the neuronal tissue. That may confuse the readers.

Paragraph starting at line 70. It could be useful for the readers to have information on the following points:

- how the social behavior of the badger also contributes to transmission of STD (groups of several animals, that exceptionally have reached 23 badgers);

- the duration of the gestation period (months after implantation);

Results

Line 126. Collected in what year?

Line 193. One semen plasma sample or one seminal plasma sample?

Line 212. displayed macroscopic lesion

Lines 214-215. What is a pale and freshform of lesion? By reddenedthe author means congestive?

Lines 201 (section 2.5) and 309 (section 3.3). Use histopathological or macroscopic/gross lesions to guide the readers.

Discussion

Had the authors into account that spermatogenesis decline in summer when comparing the values of the sperm counts for the 39 males?

Briefly comment what is known (or not) on other pathologies associated with reproductive problems in badgers.

Table 1. Column 1 is confusing.

Figure 1 and 2. Add p-value before the numbers 0.54 and 0.12, etc

References

If possible please revise and reduce the number of references that are quite excessive.

Author Response

Response to general comment:

Thank you for the helpful advice and kind words. We have fixed the errors according to the suggestions.

Reviewer Comment 1:

Keywords

Include also “latency” and “reactivation”.

Added accordingly

Abstract

Line 24 23. Add the geographical origin of the samples since it is not in the title of the manuscript.

Added “in Ireland”

Introduction

Line 37 39. The proposed fourth subfamily of mammalian herpesviruses (Deltaherpesvirinae that includes ElHV-1is not mentioned by the authors. Despite the recommendation is yet to be ratified by the International Committee on Taxonomy of viruses, the new subfamily has been largely referred in the literature.

Author response 1:

Modified as suggested. Thank you for the reminder.

Reviewer Comment 2:

Line 97 110. As it is written, it seems that the sites of latent infection include only the hematopoietic (monocytes, B cells and T cells) and not the neuronal tissue. That may confuse the readers.

Author response 2:

Modified as suggested. Thank you for pointing this out.

Reviewer Comment 3:

Paragraph starting at line 70 79. It could be useful for the readers to have information on the following points:

- how the social behavior of the badger also contributes to transmission of STD (groups of several animals, that exceptionally have reached 23 badgers);

Done (line 82-84)

- the duration of the gestation period (months after implantation);

Done (line 85)

Author response 3:

Added this information as suggested.

Reviewer Comment 4:

Results

Line 126 140. Collected in what year? Added “1996”

Line 193 218. One semen plasma sample or one seminal plasma sample? Changed to “seminal”

Line 212 237. displayed macroscopic lesion. Added accordingly

Lines 214-215 239. What is a pale and fresh form of lesion? By reddened the author means congestive? Clarified accordingly

Lines 201 227 (section 2.5) and 309 353 (section 3.3). Use histopathological or macroscopic/gross lesions to guide the readers. Rephrased accordingly

Author response 4:

Modified as suggested. Thank you for pointing these out.

Reviewer Comment 5:

Discussion

Had the authors into account that spermatogenesis decline in summer when comparing the values of the sperm counts for the 39 males?

Author response 5:

All semen samples (n= 39 males) for the sperm counts were collected in winter (i.e., during the same sampling session at the end of January / beginning of Feburary 2020). Thus, any decline in spermatogenesis after the main mating season is irrelevant in this context and does not affect the validity of our analyses. However, we agree with the reviewer that this would be an interesting area of future research.

Reviewer Comment 6:

Briefly comment what is known (or not) on other pathologies associated with reproductive problems in badgers.

Author response 6:

Added a sentence in line 316 - 317 emphasising lack of relevant studies. Thanks for the suggestion.

Reviewer Comment 7:

Table 1. Column 1 is confusing.

Figure 1 and 2. Add p-value before the numbers 0.54 and 0.12, etc

Author response 7:

Modified as suggested in Figure 1 and 2. Thanks for the suggestion.

Reviewer Comment 8:

References

If possible please revise and reduce the number of references that are quite excessive.

Author response 8:

Thank you for the suggestions, we have deleted 6 references but simultaneously had to add two more references (number 36 and 98) that are necessary to support the additional information required by the reviewers for this revision.

Additional changes to the specific points to further address the reviewers’ concerns and ambiguity:

Line 61 – 62: added EHV4 as another example of herpesvirus associated abortion

Line 99 – 102: emphasized that badger infanticide has been confirmed in England but not in Ireland

Line 111: minor rephrasing

Line 143: simplified the statistical description of the area under ROC

Line 148 - 149: minor rephrasing

Line 159: minor rephrasing

Line 166 - 167: simplified the statistical description of the area under ROC at the Table 1 description

Line 255: added citation of S3

Line 255: minor rephrasing

Line 257: replaced original Figure 4B with one that shows inclusion bodies more clearly

Line 259: minor rephrasing

Line 261: minor rephrasing

Line 271: added the range of EBV seroprevalence

Line 308: minor rephrasing

Line 376 and 381: minor rephrasing

Line 403-404: Added information on carcass preservation

Line 412 and 422: Added further information on examination procedure to determine lactation status

Line 443: minor rephrasing

Line 481: Added citation of S1

Line 495: Corrected “multivariate” to “multivariable”

Line 496 -498: Corrected information of ROC analysis

Line 505: Added description of S3

Reviewer 2 Report

In this manuscript entitled: Effects of Mustelid Gammaherpesvirus 1 (MusGHV-1) Reactivation in Badger (Meles meles) genital tracts on Reproductive Fitness, Tsai M., et al. investigated MusGHV-1 DNA prevalence in genital swabs collected from 144 wild badgers (71 males, 73 females) to predict MusGHV-1 impacts on male fertility indicators (sperm abundance and testes weight) and female fecundity (current reproductive output). In this study, PCR detection of MusGHV-1 DNA in genital swabs is considered as reactivation of MusGHV-1. Statistical analysis was used to predict reproductive fitness based on the MusGHV-1 prevalence in different groups of animals. MusGHV-1 reactivation (or detection of MusGHV-1) was found to have a negative effect on female reproduction, but not on male fertility. It was also predicted that males had a higher risk of MusGHV-1 reactivation than females, especially during the late-winter mating season. In female animals, genital MusGHV-1 reactivation differed between age classes, where 3 to 5-year-old adults had significantly lower reactivation rates than younger or older ones.

Overall, analysis of female reproductive fitness is hard to follow since results were generated from selected groups of animals that were tested for MusGHV-1. The number of animals from different age groups varies significantly. The sample size lacks power analysis. Only the “young” age group had a bigger sample size, where statistical analysis was significant. Based on the MusGHV-1 PCR test results, MusGHV-1 can be detected in both winter and spring at a similar rate, and have no significant difference between males and females, although male seems to have a higher rate (64.5-82.5%) of MusGHV-1 infection than the rate (47.5%-51.5%) in the female. The number could change if more animals were tested. Therefore, the conclusion made in this study lacks statistical power and solid data support. Another major issue with the current study is latency in all the tested animals were not evaluated, especially in those animals tested negative in genital swabs. 

It would be more interesting if this manuscript included more pathology studies related to MusGHV-1 reactivation in the reproductive tracts.  

The following are specific issues to be addressed:

  1. Were all pregnant animals with abortion scare positive for MusGHV-1? 
  2. Were genital tissues with or without lesions tested for MusGHV-1? To confirm the lesions and histopathology resulted from MusGHV-1 reactivation, it is necessary to show MusGHV-1 DNA or virion presence in the affected tissues. 
  3. It also necessary to show that aborted tissues were infected with MusGHV-1.

Author Response

Reviewer Comment 1:

Overall, analysis of female reproductive fitness is hard to follow since results were generated from selected groups of animals that were tested for MusGHV-1.

Author response 1:

Thank you for raising this potentially important concern that could indeed induce bias into any analysis. However, in our study, the animals tested for MusGHV-1 were chosen randomly where we only selected for non-road kills and freshness. Therefore, we believe that our reproductive fitness analysis using 40 females is representative for the population. The conception rate, pregnancy rate and litter size (foetus counts, scar counts and blastocyst counts) are comparable to previous studies. We concede that the abortion rate in this study is slightly higher than results reported in previous studies (Corner et al., 2015 and Rosen et al 2019), and we have added a brief discussion to this effect in the text (line 332- 334).

Reviewer Comment 2:

The number of animals from different age groups varies significantly. The sample size lacks power analysis. Only the “young” age group had a bigger sample size, where statistical analysis was significant.

Author response 2:

Thank you for pointing this out. It is very important to validate the statistical power in this study. We have added the following statistical power analysis in the method section (line 499 - 503).

We are currently working on a different (and much larger, n=251) dataset from a UK population, where we show that age has strong correlation to MusGHV-1 prevalence (quadratic term, adjusted r2=0.602, p<0.001).

We use adjusted r2 as effect size, predictor=1, significance level=0.05 to calculate the minimum size required to reach 0.8 statistical power (http://www.statskingdom.com/sample_size_regression.html), and determined that n= 8 has sufficient statistical power to reach a valid statistical result (see Table 2).

Reviewer Comment 3:

Based on the MusGHV-1 PCR test results, MusGHV-1 can be detected in both winter and spring at a similar rate, and have no significant difference between males and females, although male seems to have a higher rate (64.5-82.5%) of MusGHV-1 infection than the rate (47.5%-51.5%) in the female. The number could change if more animals were tested. Therefore, the conclusion made in this study lacks statistical power and solid data support.

Author response 3:

Thank you for raising this point. It is important here to emphasise that we are not analysing MusGHV infection rates (which are likely to be near 100%; see Sin et al., 2014 and King et al., 2004) but reactivation rates in the genital tract. Our power analysis (the results of which we now include in the methods section of the manuscript line 499 - 503) indicates, n=8 individuals per group is sufficient to ensure statistical power. However, we agree with the reviewer that a larger sample size could increase the confidence of the results representing the broader population. Nonetheless, we believe our observations are still informative and highlight interesting trends, particularly in the absence of any other information on the subject.

Reviewer Comment 4:

Another major issue with the current study is latency in all the tested animals were not evaluated, especially in those animals tested negative in genital swabs. 

Author response 4:

Thank you for raising this concern. As we detail in the introduction (line 108-109), MusGHV-1 prevalence is 98 to 100% in English and Irish badgers. Thus, we assume that also in our dataset, (almost) all sexually mature badgers will have been infected with MusGHV-1, and would thus test positive in blood PCR tests, where those animals that tested negative in our analyses of genital tract swabs were at the latent stage of their infection (i.e., no lytic infection in genital epithelial cells thus no cell-free virions present in the genital tract). We therefore endeavoured to refer to our analyses as investigating “reactivation rates” throughout the ms.

Reviewer Comment 5:

It would be more interesting if this manuscript included more pathology studies related to MusGHV-1 reactivation in the reproductive tracts.  

Author response 5:

Thank you for this suggestion. We have added the remaining histopathological results as the supplementary material 3 (S3) at line 255 to include this information while simultaneously adhering to the journal’s word limit.

Reviewer Comment 6:

The following are specific issues to be addressed:

  1. Were all pregnant animals with abortion scare positive for MusGHV-1? 

Author response 6:

The 4 females with full litter loss all tested positive, but only 1 in the 3 with partial litter loss tested positive (see line 337 - 339)

Reviewer Comment 7:

  1. Were genital tissues with or without lesions tested for MusGHV-1? To confirm the lesions and histopathology resulted from MusGHV-1 reactivation, it is necessary to show MusGHV-1 DNA or virion presence in the affected tissues. 

Author response 7:

Unfortunately, due to COVID-19-dependent lab restrictions, it was only possible for us to test only 1 tissue with lesions (Figure 3B); the result was positive, which we now make explicit in the manuscript (line 246 - 248). Because gammaherpesviruses can remain latent in epithelial cells or patrolling B cells in tissues, viral DNA may also present in tissues without lesion. Therefore, the presence of the virus DNA “in tissues” could not be used to answer if the lesion is related to MusGHV-1 reactivation, which was one of our key questions. In our opinion, active lytic infection is more reliably indicated by the inclusion bodies found during the histopathological examination, although IHC or ISH tests would be required to confirm the link unambiguously (where we were unfortunately unable to do these tests due to limited resources and particularly the COVID lab-restrictions).

Reviewer Comment 8:

  1. It also necessary to show that aborted tissues were infected with MusGHV-1.

Author response 8:

Unfortunately, we have no access to aborted foetuses as we only had carcasses from females collected after the abortion had happened. Nevertheless, we were able to collect the remains of a partially absorbed foetus from 1 out of the 3 females with partial litter loss, where the remaining 2 females had no obvious foetal tissue remains left in their uterus. Hence, we collected uterus swabs only. We decided against testing these 3 samples, however, because gammaherpesviruses link to reproductive failure events only when secondary bacterial infection is involved as reported from the murine model, cattle and humans. In addition, the virus is unlikely to pass the placental barrier (as evidenced by our results on vertical transmission detailed in line 230-231), and thus the presence of virus DNA in partially aborted foetal tissues may not be necessary to evidence MusGHV-1’s effect on abortion.

Additional changes to the specific points to further address the reviewers’ concerns and ambiguity:

Line 61 – 62: added EHV4 as another example of herpesvirus associated abortion

Line 99 – 102: emphasized that badger infanticide has been confirmed in England but not in Ireland

Line 111: minor rephrasing

Line 143: simplified the statistical description of the area under ROC

Line 148 - 149: minor rephrasing

Line 159: minor rephrasing

Line 166 - 167: simplified the statistical description of the area under ROC at the Table 1 description

Line 255: added citation of S3

Line 255: minor rephrasing

Line 257: replaced original Figure 4B with one that shows inclusion bodies more clearly

Line 259: minor rephrasing

Line 261: minor rephrasing

Line 271: added the range of EBV seroprevalence

Line 308: minor rephrasing

Line 376 and 381: minor rephrasing

Line 403-404: Added information on carcass preservation

Line 412 and 422: Added further information on examination procedure to determine lactation status

Line 443: minor rephrasing

Line 481: Added citation of S1

Line 495: Corrected “multivariate” to “multivariable”

Line 496 -498: Corrected information of ROC analysis

Line 505: Added description of S3